# Inactivation of *Escherichia coli* O157:H7 by High Hydrostatic Pressure Combined with Gas Packaging

**DOI:** 10.3390/microorganisms7060154

**Published:** 2019-05-28

**Authors:** Bing Zhou, Luyao Zhang, Xiao Wang, Peng Dong, Xiaosong Hu, Yan Zhang

**Affiliations:** 1College of Food Science and Nutritional Engineering, China Agricultural University, Beijing 100083, China; bingcau110@163.com (B.Z.); zhangly7667@163.com (L.Z.); echowang2014@163.com (X.W.); dongpeng12@hotmail.com (P.D.); huxiaos@263.net (X.H.); 2National Engineering Research Center for Fruits and Vegetables Processing, Ministry of Science and Technology, Beijing 100083, China; 3Key Laboratory of Fruits and Vegetables Processing, Ministry of Agriculture and Rural Affairs, Beijing 100083, China; 4Beijing Key Laboratory of Food Non-Thermal Processing, Beijing 100083, China

**Keywords:** high hydrostatic pressure, carbon dioxide, nitrogen, modified atmosphere packaging, *Escherichia coli*

## Abstract

The inactivation of *Escherichia coli* O157:H7 (*E. coli*) in physiological saline and lotus roots by high hydrostatic pressure (HHP) in combination with CO_2_ or N_2_ was studied. Changes in the morphology, cellular structure, and membrane permeability of the cells in physiological saline after treatments were investigated using scanning electron microscopy, transmission electron microscopy, and flow cytometry, respectively. It was shown that after HHP treatments at 150–550 MPa, CO_2_-packed *E. coli* cells had higher inactivation than the N_2_-packed and vacuum-packed cells, and no significant difference was observed in the latter two groups. Further, both the morphology and intracellular structure of CO_2_-packed *E.coli* cells were strongly destroyed by high hydrostatic pressure. However, serious damage to the intracellular structures occurred in only the N_2_-packed *E. coli* cells. During HHP treatments, the presence of CO_2_ caused more disruptions in the membrane of *E. coli* cells than in the N_2_-packed and vacuum-packed cells. These results indicate that the combined treatment of HHP and CO_2_ had a strong synergistic bactericidal effect, whereas N_2_ did not have synergistic effects with HHP. Although these two combined treatments had different effects on the inactivation of *E. coli* cells, the inactivation mechanisms might be similar. During both treatments, *E. coli* cells were inactivated by cell damage induced to the cellular structure through the membrane components and the extracellular morphology, unlike the independent HHP treatment.

## 1. Introduction

During the last decade, some mild and efficient food preservation technologies have been developed to satisfy the growing consumer demands for minimally processed and preservative-free food products [1]. Among these technologies, high hydrostatic pressure (HHP) processing technology is a new, commercially successful nonthermal technology that meets these consumer demands to some extent while retaining the sensorial and nutritional properties of freshly prepared foods [2]. Compared with conventional technologies, HHP can inactivate food spoilage and pathogenic microorganisms at room temperature, extend the shelf life of foods, and reduce damage to heat-sensitive food components and the formation of harmful food components such as heterocyclic amines (HCAs) caused by high temperature [3,4,5]. In addition, unlike thermal processing and other preservations, pressure can penetrate the entire food product to inactivate both surface and internalized microorganisms and acts instantaneously and uniformly throughout foods regardless of size, shape, and geometry [6,7].

However, two main deficiencies of HHP treatment exist that limit its commercial use in low-acid foods. One is the economic costs of the high-pressure equipment that can reach pressures up to 600 MPa or more, which are required to efficiently inactivate microorganisms in food [8]. The other is its weak inactivation of bacterial spores, the most resistant cell type known. At room temperature, high-pressure food processes have been reported to be effective in reducing or inactivating vegetative pathogens, human rotavirus, hepatitis A virus, and calicivirus in foods [5], but even ultra-high pressure levels (1000 MPa) are not effective at inactivating bacterial endospores [9]. In order to overcome these critical drawbacks of HHP, various preservative factors (hurdles) to increase or accompany the efficacy of pressure-induced inactivation of microorganisms have been thoroughly studied, including moderate temperature, pH, modified atmospheres, other nonthermal food processing methods, and antimicrobial agents such as nisin [10,11,12,13].

The combination of HHP and modified atmosphere packaging (MAP) as an effective synergistic treatment that has attracted much attention [14]. Modified atmosphere packaging (MAP) is a well-established technology that is generally used for extending the shelf-life of minimally processed foods by replacing the surrounding atmosphere of the food with a gas mixture. The gas mixture usually includes the bactericidal gas CO_2_ and the comparatively inert gases of N_2_ and O_2_. It is reported that microbial growth could be inhibited by compressed gas (CO_2_, Kr, Xe, and N_2_O) over a range of pressures (1.5 to 5.5 MPa) [15,16]. Also, combining modified atmosphere packaging (50% O_2_ + 50% CO_2_) with low high pressure (150 MPa) was investigated for shelf-life extension of carrots, and it was found that spoilage microorganisms and pathogens are more susceptible to being inactivated by HHP in the presence of gas [17]. Furthermore, due to the bacteriostatic effect of CO_2_, pressurized CO_2_, known as high-pressure carbon dioxide (HPCD), whose pressure level is less than 100 MPa, has been widely used to inactivate microorganisms in foods and has become an alternative nonthermal pasteurization technique for foods [18].

In recent years, the synergistic effect of antimicrobial gas (CO_2_) and HHP, in which the pressure level is more than 100 MPa, has been extensively reported in the literature [19,20,21]. The inactivation effect of HHP on bacteria was greatly enhanced by using a new setup to dissolve and retain the concentration of CO_2_ in fruit juices [22]. Further, low- or medium-acid fruit and vegetable juices treated with a combination of HHP and dissolved CO_2_ were also effectively preserved. More importantly, when using CO_2_ in combination with HHP, the treatment pressure could be reduced without compromising a reduction in microbial count.

However, how this treatment synergistically inactivates the valid objective microorganisms is unexplored. Furthermore, the mechanism of this synergic effect remains elusive. Thus, we determined the effect of high nitrogen (N_2_) and high carbon dioxide (CO_2_) on the inactivation effect of high pressure on *Escherichia coli* O157:H7. We investigated the morphology and cellular structures by scanning electron microscopy (SEM), transmission electron microscopy (TEM), and flow cytometry (FCM) to provide more evidence for the microbial inactivation mechanism of HHP treatment combined with gas.

## 2. Materials and Methods

### 2.1. Bacterial Strain and Culture Conditions

A stock culture of *E. coli* (CGMCC1.90), obtained from the China General Microbiological Culture Collection Center (CGMCC, Beijing, China), was maintained on nutrient agar (NA) plates (Beijing Aoboxing Biological Technology, Beijing, China). The *E. coli* inoculum was made by transferring a single colony to 20 mL of nutrient broth (NB, Beijing Aoboxing Biological Technology, Beijing, China), which was shaken at 170 rpm at 37 °C for 12 h to obtain cells in stationary phase. Cultures were inoculated by transferring a 1% (v/v) inoculum to 1.5 L nutrient broth and continuing growth under the same conditions as described above for 2.5 h to obtain cells in the middle exponential phase. Cells were harvested, washed twice in sterile physiological saline (PS, 0.85% NaCl solution, pH 6.80) and resuspended in PS. For some samples, *E. coli* cells were resuspended in sterilized lotus root sauce. Sterilized lotus root sauce was prepared by the following procedure. At first, the lotus root was homogenized by the beating machine. Then, the acquired sauce was autoclaved at 121 °C. The final number of *E. coli* cells generally ranged from 10^7^ to 10^8^ CFU per milliliter (mL).

### 2.2. Packing and HHP Processing

A 50 mL *E. coli* cell suspension was transferred to ultraviolet-sterilized polyethylene terephthalate (PET) trays (200 mL) and then conditioned under 100% CO_2_ or 100% N_2_ using a DT-6A modified atmosphere packaging machine (Dajiang machinery equipment CO., Ltd., Zhejiang, China). The other cell suspensions were transferred to sterile polyethylene plastic bags, vacuum packed, and stored at 4 °C for less than 1 h before treatment.

These samples were treated in a hydrostatic pressurization unit (HHP-750, Baotou Kefa Co., Ltd., Inner Mongolia, China) with a chamber of 30 L capacity. The pressure-transmitting fluid was water. The treatment time in this study did not include the pressurization and depressurization times. The prepared samples were placed in the pressure vessel and treated at 150, 250, 350, 450, and 550 MPa for 1 min at room temperature.

### 2.3. Determination of Viable Cells

Treated and untreated samples were serially diluted and surface plated on NA agar plates (Oxoid, Basingstoke, UK). Plates were incubated at 37 °C for 24 h, and then the colonies were enumerated. Survival was expressed as the logarithmic viability reduction log_10_ (N_i_/N_0_) with N_0_ and N_i_ representing the colony counts before and after HHP treatment, respectively. Survival counts are presented as averages ± standard deviation of three independent experiments.

### 2.4. SEM and TEM Analysis

According to previous methods [23,24], a suspension of *E. coli* cells was centrifuged at 8000× rpm for 10 min at 4 °C, the supernatant was removed, and then the pellet was resuspended in 2.5% (v/v) glutaraldehyde solution to fix for 12 h. After fixation, the cells in suspension were washed several times with 0.1 M phosphate buffer (PBS, pH 7.2) and fixed again by 1% osmium tetroxide solution (pH 7.2). After 1.5 h, the cells were washed in PBS three times and dehydrated 10 min each with a series of cold ethanol solutions (10%, 30%, 50%, 75%, and 95%). For the SEM assay, the dehydrated cells were rinsed with 50%, 70%, 90%, and 100% isoamyl acetate for 3 min each, critical point dried, and coated with gold–palladium for 60 s. Observations and photomicrographs were carried out with a Hitachi S-3400 N SEM (Hitachi Instruments Inc., Tokyo, Japan) and a JEM-1230 TEM (JEOL Japan Electronics Co., Ltd., Japan).

### 2.5. FCM Analysis

The FCM analysis of untreated cells (negative control), 75% isopropanol-treated cells (positive control), and the above-treated cells were measured as described by previous studies [23,25]. Cell suspensions were washed twice with physiological saline, resuspended in physiological saline, and adjusted to 10^8^–10^9^ CFU/mL. Then, 0.15 μL of dye mixture containing equivalent SYTO9 (Sigma-Aldrich, St. Louis, MO, USA) and propidium iodine (PI, Sigma-Aldrich, USA) were added to 50 μL of the cell suspensions and incubated for 20 min at room temperature in the dark. After that, the cells were immediately analyzed with a BD FACSCalibur flow cytometer (BD Biosciences, San Jose, CA, USA) and about 30,000 cells were collected in each run. Forward scatter and side scatter were collected, and the fluorescence signals were collected in the FL1 (green fluorescence of SYTO9 at 502 nm) and FL2 (red fluorescence of PI at 613 nm) channels [26] using Cell Quest software (Becton Dickinson, San Jose, CA, USA).

### 2.6. Statistical Analysis

All experiments were repeated at least three times. All data were statistically analyzed using Microcal Origin 8.1 (Microcal Software, Inc., Northampton, MA, USA).

## 3. Results and Discussion

### 3.1. Inactivation of E. coli in Buffer and Lotus Root Suspension

The inactivation of vacuum-packed, N_2_-packed, and CO_2_-packed *E. coli* cells subjected to high pressure at 150–550 MPa at room temperature for 1 min is shown in Figure 1a. When the pressure was at 150 MPa, there was no significant difference in inactivation levels among these cells, as indicated by less than 2 logs of inactivation for them (Figure 1a). This indicated that the synergistic effect of low high pressure and gas on the inactivation of *E. coli* cells did not occur. However, the reduction in cell counts of CO_2_-packed *E. coli* cells was about 1, 4, and 2 logs more than that of vacuum-packed and N_2_-packed cells at 250 MPa, 350 MPa, and 450 MPa, respectively (Figure 1a). Furthermore, a reduction of more than 8 log units, the detection limit, was achieved at 350 MPa for CO_2_-packed cells; the pressure for a similar inactivation effect for vacuum-packed and N_2_-packed cells was 550 MPa (Figure 1a). Thus, we conclude that the combined treatment of moderately high pressure (250–450 MPa) and CO_2_ showed a strong synergistic bactericidal effect.

Next, we sought to determine whether the similar inactivation kinetics of HHP combined with gases also exist in *E. coli* cells suspended in lotus root. Comparing the results in Figure 1a,b, the inactivation levels of samples with three packages were 1 to 3 logs in lotus root less than in buffer at 150 MPa to 450 MPa. This observation that inactivation of *E. coli* by different treatments was more extensive in the buffer than in the lotus root under all conditions may be because a complex matrix has a protective effect on bacterial inactivation compared with a buffer system [27,28]. However, the reduction of CO_2_-packed *E. coli* in lotus root was also more than in vacuum-packed and N_2_-packed cells at 250 MPa, 350 MPa, and 450 MPa (Figure 1b).

It was reported that the combined treatment with HHP and dissolved CO_2_ effectively preserved the low- or medium-acid fruit and vegetable juices compared with HHP treatment [21]. Further, a synergistic effect of the combination of HHP and CO_2_ against microorganisms inoculated in poultry sausages was found [20]. This synergistic effect may be due to the increased penetration of CO_2_ into the microorganism cells under high pressure. Therefore, we can conclude that the combined application of high pressure and different gases would have different effects on inactivating *E. coli*. The presence of CO_2_ could significantly enhance the inactivation of *E. coli* treated with high pressure, which was obvious at a moderately high pressure. Nevertheless, the presence of N_2_ did not affect the inactivation at high pressure.

### 3.2. The Morphology and Intracellular Structure Changes of E. coli Cells Treated with High Pressure Combined with Gas

From the above results, the inactivation of *E. coli* cells by high pressure was indeed affected by gases, which is obvious at 250, 350 MPa, and 450 MPa (Figure 1). Therefore, we used scanning electron microscopy to examine changes in the morphology of CO_2_-packed, N_2_-packed, and vacuum-packed *E. coli* cells in buffer exposed to these pressures. The untreated *E. coli* cells showed a morphology with a regular rod shape and smooth surface (Figure 2a). After high-pressure treatment at 250 MPa or 350 MPa, both the vacuum- and N_2_-packed *E. coli* cells had a similar morphology to the untreated samples (Figure 2b,d,e,g); however, the CO_2_-packed cells were collapsed and exhibited holes and wrinkles on the surface (Figure 2c). When the treating pressure increased to 450 MPa, N_2_-packed *E. coli* cells were still intact and exhibited similar morphology to that at 250 MPa and 350 MPa (Figure 2d,g,j), while a portion of the vacuum-packed *E. coli* cells was broken and showed cellular debris. Of note, the morphology of CO_2_-packed *E. coli* suspensions was further damaged as shown by noticeable hollows, wrinkles, or holes on their surface, and the cellular debris that was caused by cell breakdown (Figure 2f,i).

As shown in the above results, while N_2_-packed *E. coli* cells were seriously inactivated by different pressure treatments (2 to 5 log units), their morphology remained stable (Figure 1 and Figure 2d,g,j). This apparent contradiction may be because N_2_ is an inert gas and does not dissolve in the water phase, which can induce two phases at high pressure—a water phase and a gas phase [29]. Therefore, destruction of the morphology induced by HHP processing may be decreased in this complex two-phase system. In addition, compressed N_2_ may penetrate the cell to balance between the internal and external environment, which has an additional protective effect on morphology.

The effects of HHP treatment combined with gas on the intracellular structure of *E. coli* cells were assessed by transmission electron microscopy. The exposure of vacuum-packed cells to pressures ranging from 250 MPa to 450 MPa induced a slight increase in transparency within the nucleoid areas and the presence of aggregated proteins (Figure 3b,e,h). Although N_2_-packed cells showed similar changes in cellular structure to the vacuum-packed cells at 250 MPa (Figure 3b,d), the intracellular damage exhibited by N_2_-packed cells was more noticeable than with the vacuum-packed cells at 350 and 450 MPa. This was evidenced by the apparent disorganization of the genome area, the appearance of blank space in the cytoplasm and the condensation of the cytoplasmic material, and the serious intensity and frequency of protein aggregation within the cell cytoplasm (Figure 3g,j). However, the membranes of the N_2_-packed cells were organized, whereas the membranes of the vacuum-packed cells displayed winding shapes, and some of them were disrupted or detached from the cytoplasmic content (Figure 3b,d,e,g,h,j). Remarkably, the CO_2_-packed cells showed the most damage of the intracellular structures from high-pressure treatment (Figure 3c,f,i). When the treatment pressure increased, the distribution of the amorphous structures became uneven and the genome area was disorganized. Also, there were large clumps of aggregated protein in the cells. The intensity of the damage of the intracellular structures and the weak wrinkling membranes of the CO_2_-packed cells increased (Figure 3c,f,i).

Taken together, both the morphology and intracellular structure of CO_2_-packed *E. coli* cells were more strongly destroyed by high pressure. Known as the most important gas in MAP, CO_2_ can dissolve in the water phase to form carbonic acid (H_2_CO_3_) to lower the pH or to inhibit the growth of bacteria [18,30,31]. Combined with HHP, this compressed CO_2_ dissolved in the liquid state could more easily penetrate the cells [30]. This not only triggered the higher expansion on the sudden release of high pressure to cause more serious cell disruption and membrane damage (morphology damages) but also resulted in the decrease of intracellular pH and disturbance of homeostasis, as well as the extraction of microbial constituents (intracellular structure damage) [32,33]. Also, the major sites of action for CO_2_ and HHP treatment were in the cell membrane, and highly-compressed CO_2_ could more easily dissolve in and distort these regions [34]. This might explain why CO_2_-packed samples by high pressure obtained higher reductions in microbial counts and heavier destruction of the morphology and cellular structures than those treated with HHP alone. However, for N_2_-packed *E. coli* cells, the intracellular structure was seriously damaged because it is highly hydrophobic and could dissolve in and distort the cellular core to upset hydrophobic interactions in the proteins [15], but their morphology remained unchanged during HHP treatment. This may explain why the reduction in the N_2_-packed *E. coli* cells was less than that of the CO_2_-packed cells.

Thus, it seems that HHP treatment combined with CO_2_ might inactivate microorganisms by destroying the cellular structure, accompanied by cell rupture. In contrast, the N_2_-packed cells were possibly inactivated by HHP through destruction of the cellular structure.

### 3.3. Membrane Permeability of E. coli Cells

For an analysis of the membrane damage in *E. coli* cells caused by high pressure, FCM combined with PI and SYTO9 was used. Comparing the profiles of the untreated and HHP treated cells, three groups were distinguished, and then regions were constructed to enumerate events within each group using the CellQuestTM program (Figure 4). Region 1 (R1) corresponded to the living cells with intact membranes (Figure 4). Region 2 (R2) was assigned as *E. coli* cells with unknown cultivability, which were in an intermediate state between dead cells and living cells, having damaged membrane and medium membrane permeability (Figure 4). Region 3 (R3) referred to the inactivated *E. coli* cells with fully damaged membranes, exhibiting high membrane permeability (Figure 4). After a 1-min pressure-holding time at 250 MPa, the majority of the CO_2_-packed *E. coli* cells was located in R3, which were higher than that of the N_2_-packed cells and vacuum-packed cells located in R3 (Figure 4c–e). For the pressure ranging from 350 MPa to 450 MPa, although there were no significant differences in the proportion of cells in R3 among these treatments, the counts in the other two regions nearly disappeared for the combined treatment of HHP and CO_2_, while they still made up a small proportion in the N_2_-packed cells and vacuum-packed cells (Figure 4f–k). These results confirmed the synergistic effect of HHP treatment with CO_2_ on *E. coli* inactivation. 

Surprisingly, the vacuum-packed cells treated with high pressure transferred directly from R1 to R3 and did not go through R2 (Figure 4c,f,i). However, in the presence of gas, the HHP-treated cells transitioned from R1 to R2 and then to R3 as the pressure increased (Figure 4d,e,g,h,j,k). This indicated that the combined use of HHP and gases (CO_2_ and N_2_) induced intermediate cells. This may be due to the mechanism of inactivation of *E. coli* cells by HHP combined with modified atmosphere packaging, which seems to be different from that of HHP alone.

Here, we show that the morphology of the N_2_-packed *E. coli* cells did not change at different pressures but its inner structure was seriously damaged. Also, although both the morphology and intracellular structure of the CO_2_-packed *E. coli* cells were strongly destroyed by the high pressure, this damage of the morphology may have been caused by damage to the inner structure. Furthermore, both the CO_2_-packed and N_2_-packed *E. coli* cells went through an intermediate phase during the high pressure. Based on our results, we propose the following model for the inactivation of cells by combing gas and high pressure. First, the combined use of HHP and gas facilitates the penetration of gas into the *E. coli* cells, which disturbs the intracellular reactions and causes clusters of proteins and the disruption of intracellular enzymes and organelles. Second, a sudden release in pressure ruptures the cells and damages their membranes, leading to the leakage of cytoplasm components. Therefore, this combined treatment might induce a series of cellular damage at first and then act on the membrane components and the extracellular morphology, unlike the independent HHP treatment that directly ruptures the cell membrane and then leads to the loss of internal substances, which would result in bacterial death [29].

## 4. Conclusions

In this study, the effect of the combination of HHP and gases to inactivate *E. coli* has been studied. The combined use of HHP and CO_2_ had a strong synergistic effect on the inactivation of *E. coli* cells, inducing serious destruction in the morphology and the membranes and cellular structure of the cells. In contrast, the combined use of HHP and N_2_ showed a similar inactivation effect to HHP alone and destroyed only the cellular structure and the membranes of the cells. Our results provide evidence that the combination of HHP and gases has a different inactivation mechanism than that of HHP treatment. In the presence of gas, the intracellular structure of the cells was damaged at first, and then the membrane and extracellular morphology were destroyed because of the solution of gas and the release of high pressure.

## Figures and Tables

**Figure 1 microorganisms-07-00154-f001:**
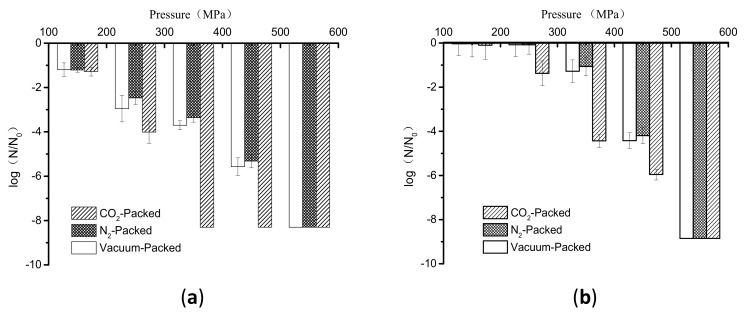
Inactivation of differently packed *E. coli* cells suspended in physiological saline by treatment with different pressures (**a**); Inactivation of differently packed *E. coli* suspended in lotus roots pulps by treatment with different pressures (**b**).

**Figure 2 microorganisms-07-00154-f002:**
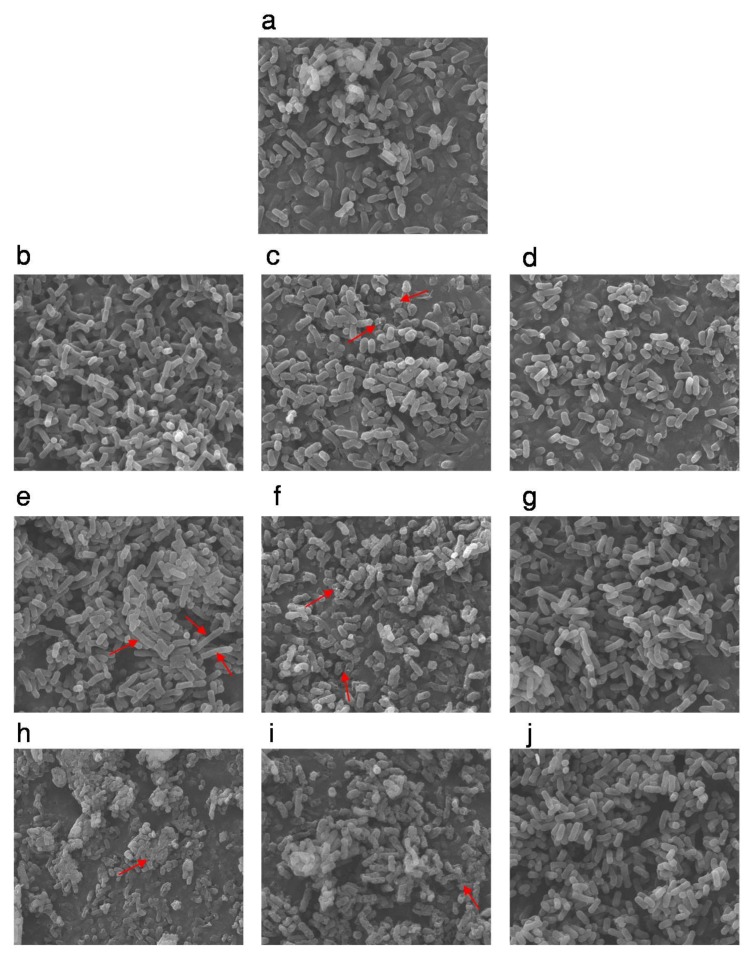
The scanning electron microscopy (SEM) images of differently packed *E. coli* before and after different pressure treatments. Untreated *E. coli* cells (**a**); vacuum-packed (**b**), CO_2_-packed (**c**), and N_2_-packed (**d**) *E. coli* cells after high hydrostatic pressure (HHP) treatment at 250 MPa: Vacuum-packed (**e**), CO_2_-packed (**f**) and N_2_-packed (**g**) *E. coli* cells after HHP treatment at 350 MPa; Vacuum-packed (**h**), CO_2_-packed (**i**) and N_2_-packed (**j**) *E. coli* cells after HHP treatment at 450 MPa. Red arrows represent the remarkable phenotypes.

**Figure 3 microorganisms-07-00154-f003:**
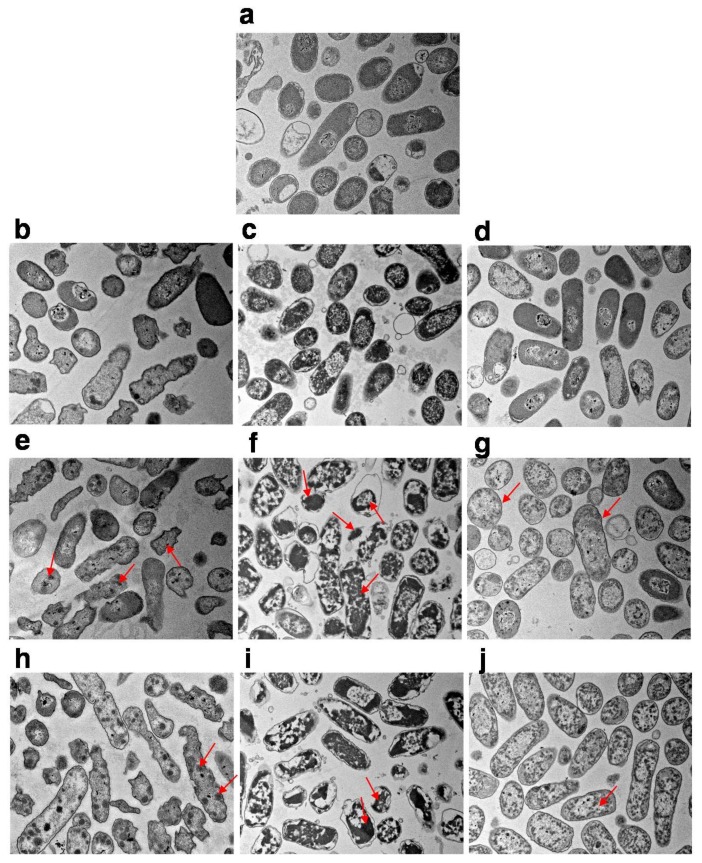
The transmission electron microscopy (TEM) images of differently packed *E. coli* before and after different pressure treatments. Untreated *E. coli* cells (**a**); vacuum-packed (**b**), CO_2_-packed (**c**), and N_2_-packed (**d**) *E. coli* cells after HHP treatment at 250 MPa: Vacuum-packed (**e**), CO_2_-packed (**f**) and N_2_-packed (**g**) *E. coli* cells after HHP treatment at 350 MPa; Vacuum-packed (**h**), CO_2_-packed (**i**), and N_2_-packed (**j**) *E. coli* cells after HHP treatment at 450 MPa. Red arrows represent the remarkable phenotypes.

**Figure 4 microorganisms-07-00154-f004:**
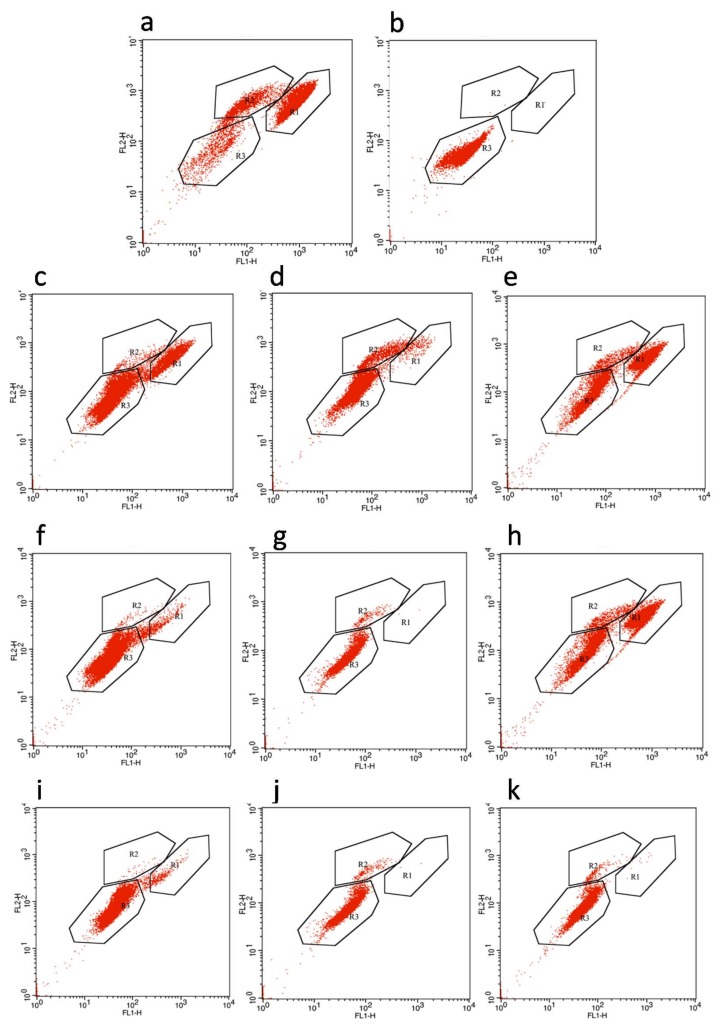
Flow cytometric analysis of *E. coli* differently packaged before and after different pressure treatments. Negative (**a**) and positive (**b**) *E. coli* cells; Vacuum-packed (**c**), CO_2_-packed (**d**), and N_2_-packed (**e**) *E. coli* cells after HHP treatment at 250 MPa: Vacuum-packed (**f**), CO_2_-packed (**g**), and N_2_-packed (**h**) *E. coli* cells after HHP treatment at 350 MPa; Vacuum-packed *E. coli* cells (**i**), CO_2_-packed (**j**) and N_2_-packed (**k**) *E. coli* cells after HHP treatment at 450 MPa.

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
