# Peer review of "Inactivation of Escherichia coli O157:H7 by High Hydrostatic Pressure Combined with Gas Packaging"

_microorganisms, 2019, doi:10.3390/microorganisms7060154_

Round 1
Reviewer 1 Report
The manuscript microorganisms-499162, presents a research on the inactivation of E.coli O157:H7 subjected to HHP combined with CO2 and N2 packaging. The topic has a high impact and the results are promising for novel application. The authors are invited to simplify and describe better the methods for the preparation of the FCM samples, determination of viability (how the samples treated were collected?) and recover of the treated samples for the SEM and TEM.
Minor comments:
48-51: add a reference;
54-58: The citation of new researches are needed because in this area there are at the moment many published studies with a high impact;
Pyatkovskyy TI, Shynkaryk MV, Mohamed HM, Yousef AE, Sastry SK. Effects of combined high pressure (HPP), pulsed electric field (PEF) and sonication treatments on inactivation of Listeria innocua. Journal of Food Engineering. 2018 Sep 1;233:49-56.
Queirós RP, Gouveia S, Saraiva JA, Lopes-da-Silva JA. Impact of pH on the high-pressure inactivation of microbial transglutaminase. Food Research International. 2019 Jan 1;115:73-82.]
65-68: the reference here is referred to carrots and not fresh salmon. Which is the study referred to?
74-78: Need a reference of the study cited. Are the authors referring to the following study?
Deng K, Serment-Moreno V, Welti-Chanes J, Paredes-Sabja D, Fuentes C, Wu X, Torres JA. Inactivation model and risk-analysis design for apple juice processing by high-pressure CO 2. Journal of food science and technology. 2018 Jan 1;55(1):258-64.
L88: specify if the strain CGMCC1.90 is pathogenic. There is no reference about O157 and why the authors used this strain instead of other serogroups or other pathogens.
L96: “sterilized lotus root sauce” - add the producer. For what reason this product has been used?
L106-106: I suggest changing in “The pressure-transmitting fluid was water”
L115: typo in the title of the paragraph.
Format: Author Contributions and Funding must be completed to meet the standard required of the journal.
Author Response
The manuscript microorganisms-499162, presents a research on the inactivation of E.coli O157:H7 subjected to HHP combined with CO2 and N2 packaging. The topic has a high impact and the results are promising for novel application. The authors are invited to simplify and describe better the methods for the preparation of the FCM samples, determination of viability (how the samples treated were collected?) and recover of the treated samples for the SEM and TEM.
Minor comments:
48-51: add a reference;
Thank you! We have added it, as shown reference 8 in the manuscript.
54-58: The citation of new researches are needed because in this area there are at the moment many published studies with a high impact;
Pyatkovskyy TI, Shynkaryk MV, Mohamed HM, Yousef AE, Sastry SK. Effects of combined high pressure (HPP), pulsed electric field (PEF) and sonication treatments on inactivation of Listeria innocua. Journal of Food Engineering. 2018 Sep 1;233:49-56.
Queirós RP, Gouveia S, Saraiva JA, Lopes-da-Silva JA. Impact of pH on the high-pressure inactivation of microbial transglutaminase. Food Research International. 2019 Jan 1;115:73-82.]
Thank you! We have cited them, as shown reference 10 and 11 in the manuscript.
65-68: the reference here is referred to carrots and not fresh salmon. Which is the study referred to?
Thank you! We have changed it.
74-78: Need a reference of the study cited. Are the authors referring to the following study?
Deng K, Serment-Moreno V, Welti-Chanes J, Paredes-Sabja D, Fuentes C, Wu X, Torres JA. Inactivation model and risk-analysis design for apple juice processing by high-pressure CO2. Journal of food science and technology. 2018 Jan 1;55(1):258-64.
Thanks! We have referred this study (reference 22).
L88: specify if the strain CGMCC1.90 is pathogenic. There is no reference about O157 and why the authors used this strain instead of other serogroups or other pathogens.
Indeed, the strain CGMCC1.90 is pathogenic, and a model organism [1,2].
L96: “sterilized lotus root sauce” - add the producer. For what reason this product has been used?
We made the sterilized lotus root sauce by ourselves. At first, the lotus root was homogenized by the beating machine. Then, the acquired sauce was autoclaved at 121℃. We have added this description in the text (Line 95-96).
L106-106: I suggest changing in “The pressure-transmitting fluid was water”
Thank you for this suggestion! We have modified it.
L115: typo in the title of the paragraph.
Thank you! We have changed it.

Reviewer 2 Report
The manuscript describes the inactivation of E. coli O157:H7 by UHPcombined with gas packaging (N2 and CO2). Moreover quite some attention is given to the visible damage to the cells using TEM and SEM. The study is a laboratory experiment that used cells in the logarithmic phase, which are not very robust cells. These cells were suspended in saline and in lotus root sauce. There was no explanation about the lotus root sauce: how prepared, is it a commercial available product, or was it used to similate presence of organic matter?
Although te set-up of the experiment used bacterial cells from the log phase, information is given about the effect of N2 and CO2. This could be useful in inactivation of pathogens present on a varieting of food products. Therefore it is a pity that the authors did not try to investigate this effect on a (naturally) contaminated food product, where cells are in another physiological state. However, maybe they will investgate this in further experiments.
It is not mentioned how long these suspensions were stored before UHP treatment (line 102/103)
I could not find results of the N0 count.
I could not find references 23 and 24 in the text (maybe I overlooked this)
In reference 26 change 2015 to bold
Author Response
The manuscript describes the inactivation of E. coli O157:H7 by UHPcombined with gas packaging (N2 and CO2). Moreover quite some attention is given to the visible damage to the cells using TEM and SEM. The study is a laboratory experiment that used cells in the logarithmic phase, which are not very robust cells. These cells were suspended in saline and in lotus root sauce. There was no explanation about the lotus root sauce: how prepared, is it a commercial available product, or was it used to similate presence of organic matter?
We made the sterilized lotus root sauce by ourselves. At first, the lotus root was homogenized by the beating machine. Then, the acquired sauce was autoclaved at 121℃. We have added this description in the text (Line 95-96).
Although te set-up of the experiment used bacterial cells from the log phase, information is given about the effect of N2 and CO2. This could be useful in inactivation of pathogens present on a varieting of food products. Therefore it is a pity that the authors did not try to investigate this effect on a (naturally) contaminated food product, where cells are in another physiological state. However, maybe they will investgate this in further experiments.
Thank you for this constructive suggestion! However, our research is aiming to investigate the inactivation mechanism of microgram by combined food treatments. So, a simple system is easier to study and detect than that in complicated systems such as contaminated food product. However, we would like to investigate this in future.
It is not mentioned how long these suspensions were stored before UHP treatment (line 102/103)
For only less than 1 h, these suspensions were stored at 4℃ before UHP treatment. We have added this information in the text (Line 104).
I could not find results of the N0 count.
We adjusted the final number of E. coli in suspension to 107 to 108 CFU per ml. as shown in line 95. Also, our results were shown in the survival ratio, expressed as the logarithmic viability reduction log10 (Ni/N0).
I could not find references 23 and 24 in the text (maybe I overlooked this)
The references 23 and 24 in the text are in line 119. Hope you can find them.
In reference 26 change 2015 to bold
Thank you! We have changed it.
1. Zhao, F.; Wang, Y.T.; An, H.R.; Hao, Y.L.; Hu, X.S.; Liao, X.J. New Insights into the Formation of Viable but Nonculturable Escherichia coli O157:H7 Induced by High-Pressure CO2. Mbio 2016, 7, doi:ARTN e00961
10.1128/mBio.00961-16.
2. Zhao, F.; Bi, X.; Hao, Y.; Liao, X. Induction of viable but nonculturable Escherichia coli O157:H7 by high pressure CO2 and its characteristics. PloS one 2013, 8, e62388, doi:10.1371/journal.pone.0062388
